

# Stress, sense of coherence and quality of life among Norwegian nurse students after a period of clinical practice

Benedicte Kleiveland[1], Gerd Karin Natvig[1] and Randi Jepsen[2,3]

[1] Department of Global Public Health and Primary Care, University of Bergen, Bergen, Hordaland, Norway
[2] Faculty of Health Studies, Sogn og Fjordane University College, Førde, Norway
[3] Nykøbing F. Hospital, Region Zealand, Nykøbing F., Denmark

## ABSTRACT

**Background.** Previous research has found that sense of coherence is significantly related to aspects of health, but studies on nurse students with a salutogenic approach are limited.

**Objectives.** To investigate (1) if nurse students' experience of stress differs among clinical practice in nursing homes and medical/surgical wards. (2) Whether sense of coherence and stress are associated with quality of life. (3) If sense of coherence acts as a moderator in the relationship between stress and quality of life.

**Participants.** Data were collected from 227 nurse students between January and April 2014.

**Methods.** Questionnaires measuring stress, sense of coherence and quality of life were completed after a period of clinical practice. Linear regression analyses were used to measure associations between stress, and sense of coherence respectively, and quality of life.

**Results.** The results showed that 33.92% of the students experienced moderate or high levels of stress, and there was significantly more stress in hospital wards compared to nursing homes ($p = 0.027$). Sense of coherence was positively associated with quality of life in the simple and multiple regression analyses ($p < 0.01$). Stress was negatively associated with quality of life in the simple regression analysis ($p < 0.01$), but not in the multiple analyses when sense of coherence was included. However, when we included an interaction term, stress was no longer associated with quality of life and sense of coherence appeared to be a significant moderator in the relationship between stress and quality of life ($p = 0.015$). Thus, a negative association was seen among students with the lowest levels of sense of coherence.

**Conclusion.** These findings suggest that sense of coherence could be seen as a resource that nurse educators can build upon when supporting students in coping with stress.

Corresponding author
Benedicte Kleiveland,
ben-kle@hotmail.com

## INTRODUCTION

Norwegian nursing education is part of higher education provided by the universities and university colleges. The purpose of the education is twofold: qualifications for nurse authorisation and for a Bachelor's degree. Clinical practice is a vital part of Norwegian nurse students' education with 90 out of 180 European Credit Transfer System credits included in the three-year study. Clinical training includes medical and surgical wards, mental health services, nursing homes, and home care. The duration of clinical practice lasts from one to 16 weeks, and students receive regular supervision, follow-up and assessment from teachers and staff (*Norwegian Ministry of Education and Research, 2008*). However, several studies indicate that clinical training is in an environment that may cause students to experience high levels of stress and anxiety (*Moscaritolo, 2009*; *Sharif & Masoumi, 2005*; *Timmins & Kaliszer, 2002*).

## BACKGROUND

It is assumed that stress can cause general health complaints like headache (*Nash & Thebarge, 2006*), abdominal pains and anxiety (*White & Farrell, 2006*). Unpredictable or incomprehensible life situations are important sources for stress (*Lazarus & Folkman, 1984*). *Horowitz, Wilner & Alvarez (1979)* observed that the most commonly reported response to a stressful event was intrusive and avoidant thoughts. Intrusion includes unbidden thoughts, feelings and dreams. Avoidance is characterized by concisely denial of meaning and consequences related to a stressful event. *Antonovsky (1987)* indicated that the concept of sense of coherence (SOC) might influence if people would view stressful events as chaotic and incomprehensible or coherent and comprehensible. SOC arrives from the salutogenic approach, which is the search for causes for health, rather than causes for diseases. SOC consists of three core components: comprehensibility, which is a person's perception that the internal or external environment is structured, predictable and consistent; manageability which is the belief that resources are available for dealing with problems; and meaningfulness which is a person's perception that life events have meaning and are worthy investing energy in *Antonovsky (1979)* and *Antonovsky (1987)*.

In the process linking stressful situations to health, it is possible to assume that increased levels of stress might be related to decreased levels of quality of life (QoL). Research on stress in nurse students has found stress to interfere with learning and contribute to poor mental health (*Melo, Williams & Ross, 2010*). Stress has also been a significant factor for students leaving nursing schools (*Lindop, 1987*). Despite this, as far as we know, no research has been performed to examine associations between stress and QoL in nurse students. However, the literature has found a relationship between high stress and decrement in QoL among hospital employees (*Mosadeghrad, Ferlie & Rosenberg, 2011*), teachers (*Yang et al., 2009*), and haemodialysis patients (*Shafipour et al., 2010*).

It has been suggested that SOC may influence individuals' QoL, and this relationship has been investigated in several clinical and non-clinical studies (*Eriksson & Lindstrom, 2007*). *Moksnes, Lohre & Espnes (2013)* investigated the association between SOC and QoL in adolescents and found a strong and positive association. In addition, findings from studies

in various samples such as patients with coronary heart disease (*Motzer et al., 2003*), older persons (*Ekman, Fagerberg & Lundman, 2002*; *Nesbitt & Heidrich, 2000*), patients with chronic illness (*Delgado, 2007*), educators (*Harri, 1998*) and care givers (*Ekwall, Sivberg & Hallberg, 2007*) indicate a direct or indirect positive relationship between SOC and QoL.

According to *Antonovsky (1987)*, persons with a strong SOC have general confidence that resources are available to meet demands caused by stressful events. This could mean that SOC has the ability to modify the effect stress has on QoL. Antonovsky suggested that a strong SOC would determine if the outcome of a stressful event would be harmful, neutral or salutary. A moderator is a variable that affects the directions and/or strength of the conditions between an independent variable and a dependent variable (*Baron & Kenny, 1986*). The idea of SOC as a moderating variable has been examined in several studies with mixed results. *Drageset et al. (2009)* did not find indications of a moderating effect of SOC on the relationship between social support and health related QoL. However, *Eriksson & Lindstrom (2007)* found that SOC had a moderating effect on health in stressful situations. That is, persons with a high SOC were coping better with stress than persons with a low SOC. In addition, *Albertsen, Nielsen & Borg (2001)* suggested that SOC was a moderator between work environment and stress symptoms, and *Torsheim, Aaroe & Wold (2001)* found a marginal support for SOC as a moderator between school-related stress and health complaints.

To gain more knowledge on nurse students after a period of clinical practice, the aim of this study was to examine the relationship between stress, SOC and QoL in nurse students in a Bachelor's program. We hypothesized that:

1. Levels of stress would differ between practice in nursing homes and medical/surgical hospital wards.
2. There would be an association between stress and QoL.
3. There would be an association between SOC and QoL.
4. SOC would have a moderating effect in the relationship between stress and QoL.

## METHODS

### Design and data collection

The study had a cross-sectional design. The data collection was conducted at a university college in one of the largest cities in Norway between January and April 2014. The clinical practice periods in nursing homes lasted for 8 weeks (first year students), and the clinical practice in surgical and medical hospital wards lasted for 12 weeks each (second year students). The students were given supervision by both preceptors and university teachers during their clinical practice. All students who had completed practice either in medical/surgical wards or in nursing homes within the last week were asked to participate in the study.

### Measures

All data were self-reported and the questionnaires were administered in class by lectures.

## Demographic data

We collected data on year of birth, gender, marital status, any children, and if the participants were part time or full time students.

## Impact of Event Scale

The Impact of Event Scale (IES) is a 15-item questionnaire, which is developed by *Horowitz, Wilner & Alvarez (1979)* and assesses subjective distress related to a self-defined stressful event. The questionnaire was composed of commonly reported experiences of intrusion and avoidance. The students were asked to describe and rate a stressful event that occurred in the clinical environment. The specific event and the date of its occurrence were recorded on the top of the page. The first part of the questionnaire concerns intrusive memories of the event, such as nightmares and penetrating thoughts during the last week. The second part concerns the extent of avoidance, such as trying to push away memories during the last week. The scoring alternatives are 0 (never), 1 (rarely), 3 (sometimes), and 5 (often). The summed score ranges from 0 to 75. Scores from 0 to 8 are considered subclinical levels, 9–25 mild levels, 26–43 moderate levels and 44+ severe levels of stress. A cut-off point of 26 has been suggested to define a moderate or severe level of stress (*Horowitz, Wilner & Alvarez, 1979*). *Joseph (2000)* and *Sundin & Horowitz (2002)* reviewed several studies that reported the validity and reliability of the IES-15 as acceptable. The Norwegian version has shown good psychometric properties in a study on medical students (*Eid, Thayer & Johnsen, 1999*).

## Quality of life scale

The Quality of life scale (QOLS) is a 16-item domain-specific instrument. It measures an individual's overall satisfaction with life in various domains. These are: material comfort, health, independence, recreation and relationship with others (*Burckhardt, Archenholtz & Bjelle, 1992*; *Burckhardt, Archenholtz & Bjelle, 1993*; *Burckhardt, Clark & Bennett, 1993*; *Burckhardt et al., 1989*). The response options range from "very satisfied" to "very dissatisfied." The summed score ranges from 16 to 112, with a higher score indicating a better QoL.

QOLS-16 has demonstrated acceptable reliability and validity in studies across groups and cultures (*Burckhardt & Anderson, 2003*; *Burckhardt et al., 2003*). The Norwegian version has been validated by *Wahl et al. (1998)*.

## Sense of coherence

Antonovsky developed the SOC questionnaire for empirical testing of his theory. He developed two versions: the original that comprises 29 items, and a short-form that comprises 13 items. In both versions, the items are scored on a 7 points Likert scale with response alternatives from 1 ("not at all") to 7 ("very often"). In the present study we used the SOC-13. SOC-13 includes the components comprehensibility, manageability and meaningfulness (*Antonovsky, 1987*). The total score ranges from 13 to 91 points. A high score indicates a strong SOC. A systematic review of Antonovsky's SOC scale concluded that the validity and reliability of the 13-item scale are acceptable (*Eriksson & Lindstrom, 2005*).

The SOC questionnaire has been used in cross-sectional clinical and non-clinical studies (*Blom et al., 2010*).

## Ethical considerations

The Norwegian Data Protection Official for Research, the dean and the student council approved this study. In writing, the students were informed that participation in the study was voluntary and anonymous. Completion of the questionnaire was considered as consent to participation. Completed questionnaires were placed in sealed envelopes without identification and returned to the researcher.

## Data analysis

The data were analysed using SPSS version 21. Descriptive statistics were used to analyse the socio-demographic data, the QOLS-16, IES-15 and SOC-13. We divided the material into three categories: medical ward, surgical ward and nursing homes. The mean and standard deviation (SD) of the sum scores were calculated for SOC-13, IES-15 and QOLS-16. Differences between average sum scores for the three categories were examined using One-Way ANOVA test. Cronbach's alpha was used to assess the internal consistency of the scales.

We used unadjusted (Model 1) and adjusted (Model 2) multiple linear regression analysis to test the relationship between SOC-13 and IES-15 respectively, and QOLS-16. For the adjusted analyses, we dichotomized the socio-demographic control variables as: Gender: 1 = women, 0 = men; material status: 1 = married/cohabiting, 0 = Boyfriend/girlfriend/single; children: 1 = no children, 0 = children; studies: 1 = part-time, 0 = full-time; place of practice: 1 = hospital, medical/surgical hospital wards, 0 = nursing homes; and age: 1 = under or 25, 0 = over 25.

The interaction effect between SOC-13, IES-15, and QOLS-16 was tested in the regression analysis. The level of significance was set to $p \leq 0.05$. Regression analysis is known to be unsatisfactory and unstable when including an interaction term with raw variables (*Baron & Kenny, 1986*). Therefore, the independent and the moderating variables were centered before they were entered into the regression analyses.

## RESULTS

A total of 308 students met the inclusion criteria, and 232 gave consent to participation in the study. We used data from the 227 participants who completed more than 50% of the items, yielding a response rate of 73.7%.

Missing values were replaced with variable means (*Ware, Kosinski & Gandek, 2000*). The percentage of missing items on IES-15 was 0.4% (item 8). For the QOLS-16, missing items amounted to 0.4% (item 13 and 14), 1.3% (item 8, 11 and 12), 4.0% (item 5), and 13.7% (item 4). On the SOC-13 scale, there were no missing items.

Table 1 presents the socio-demographic data. The majority (82.4%) was women and the mean age was 27.4 (min 20 max 59) years old. Almost half of the participants (44.5%) reported that they were single, the majority (73.6%) did not have children, and 75.3% were full-time students.

**Table 1 Descriptive statistics of the participants. Nominal independent variables ($n = 227$).**

| Socio-demographic variables | | $n$ (%) |
|---|---|---|
| **Gender** | Male | 40 (17.6%) |
| | Female | 187 (82.4%) |
| **Age** | 19–26 | 147 (64.8%) |
| | 27–34 | 42 (18.5%) |
| | 35–42 | 20 (8.8%) |
| | 42+ | 18 (7.9%) |
| **Education** | Full-time students | 171 (75.3%) |
| | Part-time students | 56 (24.7%) |
| **Any children** | Yes | 60 (26.4%) |
| | No | 167 (73.6%) |
| **Marital status** | Married | 39 (17.2%) |
| | Cohabitant | 49 (21.6%) |
| | Boyfriend/girlfriend | 38 (16.7%) |
| | Single | 101 (44.5%) |

**Table 2 Distribution scores for SOC-13, QOLS-16 and IES-15.**

| | Whole sample ($N = 227$) | Nursing homes ($N = 109$) | Medical wards ($N = 58$) | Surgical wards ($N = 60$) | $p$-values[*] |
|---|---|---|---|---|---|
| **SOC-13:** | | | | | |
| Mean (SD) | 61.87 (10.51) | 63.25 (10.56) | 60.75 (9.95) | 60.45 (10.84) | 0.165 |
| **QOLS-16:** | | | | | |
| Mean (SD) | 83.68 (10.92) | 85.03 (10.55) | 82.95 (11.32) | 81.91 (11.03) | 0.604 |
| **IES-15:** | | | | | |
| Mean (SD) | 20.48 (14.84) | 17.74 (13.38) | 22.97 (16.07) | 23.05 (15.30) | 0.027 |
| $\geq$26.%[a] | 33.92% | 26.61% | 37.93% | 43.33% | |

**Notes.**

[*] $p$-values measured with one-way ANOVA, significant at the level of 0.05.

[a] Cut-off point: displays the prevalence of scores over 26 in per cents.

As shown in Table 2, the mean IES-15 was 20.48 with a significant difference between the places of practice. We found less stress in students who had been in practice in nursing homes compared to those who came from medical or surgical wards. 33.92% of the students had a higher score than 26, which indicates a moderate or severe range of stress (*Hutching & Devilly, 1999*). The mean QOLS-16 was 83.68, and the mean SOC-13 score was 61.87.

As seen in Table 3, we found that QOLS-16 was not significantly associated with the socio-demographic variables in the unadjusted and adjusted regression analyses. However, QOLS-16 was significantly associated with IES-15 and SOC-13. After adjusting for the dichotomized variables, only SOC-13 gave a significant contribution to the model.

**Table 3** Multiple linear regression analyses for the QOLS-16 for 227 nurse students in Norway.

|  | Model 1, unadjusted<br>b 95% CI $p^*$ | Model 2, adjusted[a]<br>b 95% CI $p^*$ |
| --- | --- | --- |
| *Gender* | | |
| Women | −1.79 (−6.07, 2.50) | −1.68 (−4.69, 2.36) |
| Men | 0 | 0 |
| *Marital status* | | |
| Married/cohabiting | 0.59 (−2,35, 3.53) | −1.37 (−4.18, 1.43) |
| Boyfriend-girlfriend/single | 0 | 0 |
| *Children* | | |
| No children | −1.92 (−5.16, 1.31) | −2.41 (−6.51, 1.68) |
| Children | 0 | 0 |
| *Education* | | |
| Part-time | −0.78 (−2.54, 4.1) | −3.72 (−8.35, 0.92) |
| Full-time | 0 | 0 |
| *Place of practice* | | |
| Medical/surgical ward | −2.58 (−5.43, 0.26) | −1.49 (−4.31, 1.33) |
| Nursing homes | 0 | 0 |
| *Age* | | |
| Under 25 | −0.69 (−3.62, 2.25) | −1.42 (−5.01, 2.17) |
| Over 25 | 0 | 0 |
| **SOC** | 0.63 (0.52, 0.74) <.001 | 0.62 (0.50, 0.74) <.001 |
| **IES-15** | −0.19 (−0.28, −0.09) <.001 | −0.12 (−0.99, 0.75) |

**Notes.**

$^*$ $p \geq 0.05$ was not reported.

CI, confidence interval; b, regression coefficient.

[a] Adjusted for gender, marital status, children, education, place of practice and age.

As seen in Fig. 1, we found that SOC-13 significantly interacted the relationship between IES-15 and QOLS-16 ($p = 0.015$). This analysis was also performed while adjusting for the dichotomized variables, giving a significant result (results not shown).

Cronbachs alpha was calculated to 0.90 for IES-15 and 0.84 for both QOLS-16 and SOC-13.

## DISCUSSION

This study on nurse students found that the participants reported clinical practice as stressful. Nearly one third of the students were considered to have moderate or high range of stress related to a self-defined situation in clinical practice. *Horowitz, Wilner & Alvarez (1979)* suggested that intrusive thoughts and avoidance are regarded as problematic when they become prolonged or excessive. The result is consistent with previous research that demonstrates stress in nurse students. *Blomberg et al. (2014)* found that a high level of stress was frequent among nurse students during their clinical practice. *Sharif & Masoumi (2005)* also found that nurse students reported clinical practice as stressful.

We found that clinical practice in surgical and medical hospital wards was significantly more stressful than clinical practice in nursing homes, which corresponds with our

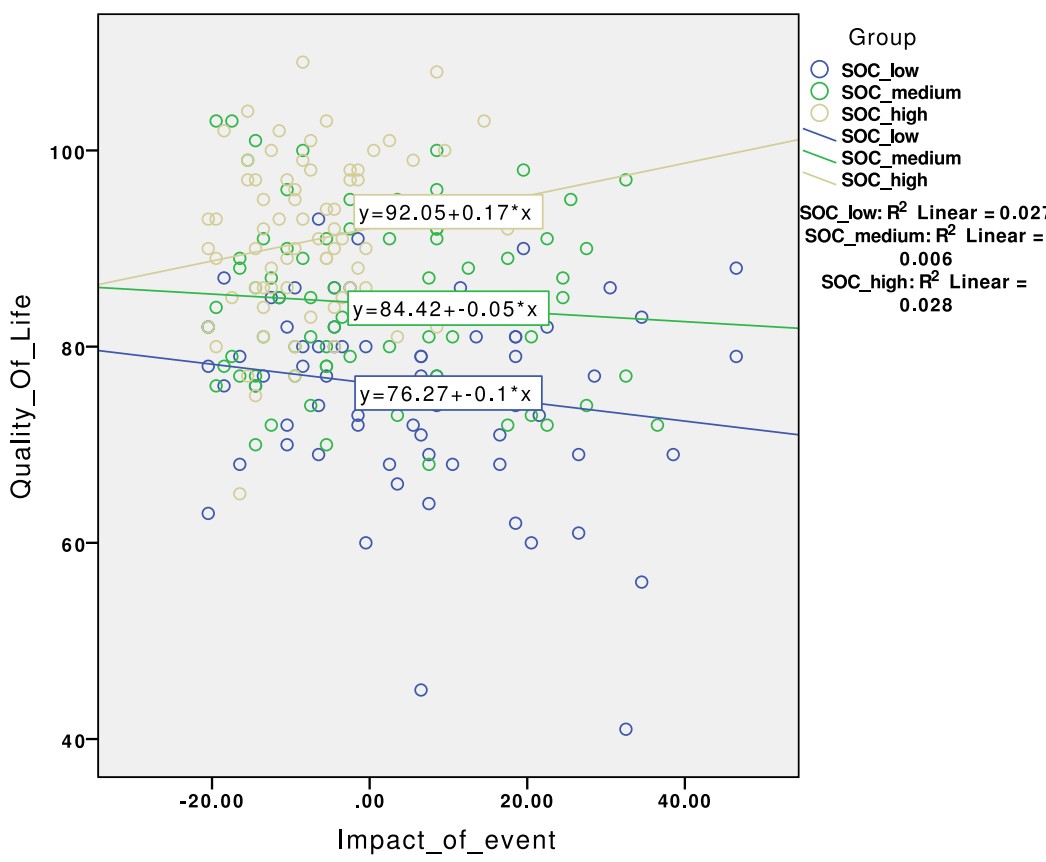

**Figure 1** Scatter diagram of sense of coherence as a moderator between stress and quality of life.

hypothesis that the level of stress differs between places of practice. Our result corresponds with *Blomberg et al. (2014)* who reported higher levels of stress in hospital wards than in community settings. This could be explained by previous findings on sources of stress that may be more frequent in medical or surgical wards, such as feeling uncertain or feeling a lack of competence and knowledge (*Burnard et al., 2008*; *Morrell & Ridgway, 2014*; *Zupiria Gorostidi et al., 2007*). In addition, previous research has suggested that hospital wards, which are overcrowded with patients, increase the levels of stress in nurse students (*Blomberg et al., 2014*). *Sveinsdottir, Biering & Ramel (2006)* suggested that nurses working in medical or surgical wards experienced more stress than nurses working outside hospitals. The hospital nurses had more direct patient care, less opportunity to take lunch breaks in the appointed time, and there were greater staff shortages. These professional stressors may be transferred to the students' work environment. In addition, increasing demands and expectations throughout the education may also contribute to the higher levels of stress in the second year students.

The hypothesis of a positive association between SOC and QoL was supported in the unadjusted and adjusted analyse. This finding suggests that students who have a high SOC could have a better QoL. Several studies have shown that SOC is correlated with QoL

(*Ekman, Fagerberg & Lundman, 2002*; *Moksnes, Lohre & Espnes, 2013*; *Motzer & Stewart, 1996*). This finding supports that SOC is a salutary resource for nurse students' QoL.

We found that stress was negatively associated with QoL in the unadjusted analysis only. This finding suggests that stress might have a negative association with nurse students QoL. Negative associations between stress and QoL have been found in previous studies (*Mosadeghrad, Ferlie & Rosenberg, 2011*; *Shafipour et al., 2010*; *Yang et al., 2009*). However, when we included SOC as an interaction term, stress was no longer associated with QoL. The previous studies are therefore not directly comparable, as they did not use SOC as an interaction term. This finding also confirmed that SOC may have a moderating role in the relationship between stress and QoL and suggests that students with a high SOC were more inclined to view demands in clinical practice as less threatening to their QoL. They might see the demands as more comprehensible, meaningful and predictable. This result is in accordance with Antonovsky's theory (*Antonovsky, 1987*).

## Study limitations

The main limitation of this study was that it only included students from one Norwegian university college. Therefore, it may not be representative for other nurse students. On the other hand, the Norwegian society is rather homogenous and the nursing education is well regulated (*Norwegian Ministry of Education and Research, 2008*). As the study had a cross-sectional design, causal relationship cannot be inferred. Other external factors than those we investigated could have contributed to the result and led to biases.

## Conclusion and implication for practice

Little research has been performed on the association between stress and SOC respectively and QoL in nurse students after a period of clinical practice. We hypothesized that levels of stress would differ according to clinical practice, that SOC and stress were associated with quality of life, and that SOC could act as a moderator in the relationship between stress and quality of life. These hypotheses were confirmed and the study gives support to the theory that Antonovsky formulated, that individuals with a high SOC might have a salutary outcome of stress. These findings suggest that SOC could be seen as a resource that nurse educators can build upon when supporting students in coping with stress.

## ACKNOWLEDGEMENTS

We would like to thank the nurse students who participated in this study. Thanks to the university college and lecturers for the help with data collection. They generously gave up some teaching time in order to allow the students answer the questionnaires.

### Funding

The authors declare there was no funding for this work.

## Competing Interests

The authors declare there are no competing interests.

## Author Contributions

- Benedicte Kleiveland conceived and designed the experiments, performed the experiments, analyzed the data, wrote the paper, prepared figures and/or tables, reviewed drafts of the paper.
- Gerd Karin Natvig and Randi Jepsen contributed reagents/materials/analysis tools, reviewed drafts of the paper.

## Human Ethics

The following information was supplied relating to ethical approvals (i.e., approving body and any reference numbers):

1. The Norwegian Data Protection Official for Research
2. Reference number 35907.

## Supplemental Information

Supplemental information for this article can be found online at http://dx.doi.org/10.7717/peerj.1286#supplemental-information.

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
