# Peer review of "Stress, sense of coherence and quality of life among Norwegian nurse students after a period of clinical practice"

_PeerJ, doi:10.7717/peerj.1286_

## Round 0.1 · original submission · Minor Revisions

Thank you for this interesting manuscript on stress, quality of life and sense of coherence in nursing students. It would be helpful to have more of an explanation as to why the Impact Event Scale was chosen to measure stress in this population. In addition, it seems that there are a few other limitations to add to your study, that is, were there other reasons for stress in this group of participants, did it matter that the students who were in the nursing home setting were 1st year students, while the students in the medical-surgical acute care were 2nd year students? As noted in your discussion those in the medical surgical wards may have feelings of uncertainty or feel a lack of competence, but those with less experience were placed in the less stressful environment of the nursing home.

Reviewer 1 ·

Basic reporting

The manuscript addresses the phenomenon of stress levels in professional nursing education and the relationship to Sense of Coherence, a construct for Antonovsky’s Salutogenic Model of Health, and quality of life. This relationship has been previously studies, but not in healthy populations. Therefore the study fills a gap in the extant knowledge in the area. The research hypotheses asked about the relationship between Quality of Life and Sense of Coherence and about the level of stress associated with different practice sites.

Experimental design

The methodology selected for the study is a quasi-experimental, cross-sectional, self-reported survey of nursing students in a northern European country. Three variables were examined. Stress was operationalized by the Impact Event Scale (IES) developed by Horowitz and others in 1979. This survey has acceptable reliability, but there are more specific stress scales available and the choice of this measure is not fully explained. Quality of life was operationalized by the Quality of Life survey (QOLS) (Burckhardt et al, 1992). It is domain specific and useful for healthy persons, making it a better choice than some Quality of Life measures that are designed for use with persons with chronic health conditions. Sense of coherence was operationalized by the SOC 13 (short form) developed by Antonovsky. This has been demonstrated to be as reliable as the longer 29 item questionnaire.
The use of professional nursing students as a sample representative for high stress in a non-clinical population makes much sense as there is documentation of high stress in this and other professional healthcare provider students related to the challenges of the preparation for and the nature of the professional service, as well as the difficulty in coordinating and balancing the educational demands with a home and family life. Practice site differentiation identified nursing homes as a lower stress area than acute care facilities.

Validity of the findings

The statistical analysis produced two regression models; one was adjusted to allow for instability in using raw data in a linear regression, but the final models were quite similar. The inverse relationship between stress and SOC was again noted, as was the negative impact of stress on life quality. Both were significant to the level of 0.001. Sense of coherence was found to have a moderating effect on stress as it affected quality of life. The impact of other demographic variables did not rise to significant levels.
The study confirmed existing knowledge on stress, Sense of coherence and quality of life using a healthy population which had not been studies in depth previously. A limitation of the study could be the small number of demographic variables included. The inclusion of demographic variables such as the number of children, socioeconomic status, and employment status of the student could impact stress levels and might have altered the results.

Reviewer 2 ·

Basic reporting

No Comments

Experimental design

1. More detailed information about design of study is needed, for example data were collected from nursing students between Jan and April 2014. The study had a cross-sectional design?
2. Inclusion criteria is not clear
3. Reliability and validity of the instruments should be described more detailed.
4. More detailed information about the investigated Instrument is needed.
5. Having permission of the developer (instrument developer) for using the instrument?
6. Hypotheses are not supported by references

Validity of the findings

In discussion, results are not supported with related studies for example several studies have shown that SOC is correlated with QoL, all references related to patients (L. 195).

---

## Round 0.2 · accepted · Accept

Thank you for your revised manuscript on stress, quality of life and sense of coherence in Norwegian nursing students. Your edits have answered our previous inquiries.

Reviewer 1 ·

Basic reporting

The rewritten submission is clearly written and conforms to professional standards.

Experimental design

The research design is clearly described and the question relevant and meaningful. The added information on instrument reliability clearly demonstrates the rigor of the study.

Validity of the findings

Data reporting is well done.

Additional comments

The rewrite is very well done.

Reviewer 2 ·

Basic reporting

No Comments

Experimental design

No Comments

Validity of the findings

No Comments

Additional comments

There are some questions that better answered by authors:
1) More information would be useful on the translation procedures for the scales used.

2) Inclusion criteria is not clear.

·

Basic reporting

The title of the article is “Stress, sense of coherence and quality of life among Norwegian nurse students after a period of clinical practice”. All the aspects mentioned in the title are being researched and discussed in the article.

Experimental design

I was able to read the previous reviewers comments as well as the answers given by the authors to questions posed to them. I believe they answered the questions well and added more info to the clarity of the article.

Validity of the findings

The authors answered the previous reviewers’ questions. A suggestion: expand on the comment under the heading “Conclusion and implication for practice” regarding “… nurse educators can build upon when supporting students in coping with stress”. I suggest being more specific by naming one or two ways the nurse educators can support students. This will also help towards a stronger conclusion to the article.

Additional comments

The authors used extensive references. This topic will add to the knowledge regarding the education of nursing students.